# Reverse Engineering Analysis of Optical Properties of (Ti,Cu)Ox Gradient Thin Film Coating

Jarosław Domaradzki *[ID], Michał Mazur [ID], Damian Wojcieszak [ID], Artur Wiatrowski [ID], Ewa Mańkowska and Paweł Chodasewicz

Faculty of Electronics, Photonics and Microsystems, Wroclaw University of Science and Technology, Janiszewskiego 11/17, 50-372 Wroclaw, Poland; michal.mazur@pwr.edu.pl (M.M.); damian.wojcieszak@pwr.edu.pl (D.W.); artur.wiatrowski@pwr.edu.pl (A.W.); ewa.mankowska@pwr.edu.pl (E.M.); pawel.chodasewicz@pwr.edu.pl (P.C.)
* Correspondence: jaroslaw.domaradzki@pwr.edu.pl

**Abstract:** Analysis of the optical properties of a gradient (Ti,Cu)Ox thin film is presented in this paper. The thin film was prepared using reactive co-sputtering of Ti and Cu targets. The desired elemental concentration profiles of Cu and Ti versus the thin film thickness were obtained by changing the power delivered to the magnetron equipped with Cu, while the powering of the magnetron equipped with the Ti target was maintained at a constant level during the film deposition. Optical properties were analysed using the reverse engineering method, based on simultaneously measured optical transmittance and reflectance. Detailed microstructure analysis performed using transmission electron microscopy investigations revealed that the thin film consisted of at least four areas with different structural properties. Finding a satisfying fit of theoretical to experimental data required taking into account the heterogeneity in the material composition and microstructure in relation to the depth in the prepared gradient thin film. On the basis of the built equivalent layer stack model, the composition profile and porosity at the cross-section of the prepared gradient film were evaluated, which agreed well with the obtained elemental and microscopy studies.

**Keywords:** gradient thin film coating; optical properties; reverse engineering; CuO; Cu$_2$O; TiO$_2$; magnetron sputtering; microstructure

## 1. Introduction

Optical thin films based on materials with a graded refractive index have been in use for many years. Such thin films may allow one to obtain the optical properties of different types of coatings with much better performance compared to coatings manufactured on the basis of the classic multilayer system [1]. One of the examples of the use of such graded thin films is an antireflective coating [2,3], where the refractive index value changes from large on the substrate to much smaller on the surrounding area, according to a desired profile. The advantages of such antireflective coating in the form of a graded index film compared to the corresponding layered system are lower light scattering losses due to the lack of interfaces. Another example where a graded index film may find practical application is as a rugate filter [4,5].

The challenging task in the subject of graded index coating is each step of its fabrication, from its design through the technology of its preparation, to the appropriate characterisation of its properties [6]. The most obvious method for preparing a gradient coating is to design a stack of layers in which the refractive index of individual sublayers varies by a small amount. However, it is important that the thicknesses of the individual sublayers are not greater than 10 nm. At this thickness, the interference effects that occur at the interfaces of particular sublayers are negligible in the wavelength range of the visible and infrared bands [1]. Each of such individual sublayers must be characterised by a certain calculated refractive index value. However, in practice, the manufacturing of sublayers with any

desired refractive index value is difficult. One of the effective ways to solve this problem is to prepare mixtures of different materials, such as those with extremely different refractive index values, and mix those together. The composite thus produced would then have an effective refractive index of an intermediate value, depending on the mutual ratio of the constituent materials. In addition, the refractive index may change versus the graded thin film thickness according to a different designed profile—e.g., linear, parabolic, exponential, Gaussian, etc. [2,3]. The values of the refractive indices for each particular sublayer can be determined on the basis of the appropriate relation. In the simplest case, the gradient optical thin film could be a mixture of two components whose relative proportion varies as a function of the thin film thickness. For example, for a linear profile, the refractive index value for each particular sublayer, from the substrate to the surrounding area, can be derived using a simple relation:

$$n = n_i + (n_s - n_i)x \tag{1}$$

where, $n_i$—refractive index of the surroundings, $n_s$—refractive index of the substrate, and $x$—parameter that determines the ratio of depth for each particular sublayer to the total thickness of the gradient coating [7]. More complex profiles are also possible. Some examples can be found in [8].

Analysis of the optical properties of optical thin film coatings is often based on measurement of the light that is reflected from or transmitted through the coating vs. the light wavelength in a broad range. Reverse engineering is then applied that relies on finding a model that allows calculation of the theoretical curves that match the experimental results. For single, homogeneous oxide films, the model assumes an identical complex refractive index throughout the whole volume of the film. In the case of more complex materials, e.g., composite materials, the optical model often applied for analysis of their properties is based on the effective medium approximation (EMA). EMA assumes that the properties (dielectric function) of the material can be determined with an equivalent (effective) parameter, the volume fraction factor—$f$. The original EMA model elaborated by Maxwell-Garnet [9] was further developed by Bruggemann [10] and is most often used in the following form [11,12]:

$$(1-f)\frac{\varepsilon_m - \varepsilon_{eff}}{\varepsilon_m + 2\varepsilon_{eff}} + f\frac{\varepsilon_p - \varepsilon_{eff}}{\varepsilon_p + 2\varepsilon_{eff}} = 0, \tag{2}$$

where: $\varepsilon_{eff}$—effective relative permittivity, $\varepsilon_m$—matrix relative permittivity, $\varepsilon_p$—particle relative permittivity, and $f$—fraction factor (volume fraction).

In spite of the fact that many studies on analysis methods of optical properties on homogenous and inhomogenous films have been published so far, examples on the complex analysis of inhomogenous films based on the measurement of optical characteristics are still rather rare. Some practical considerations can be found in, e.g., [13] where the authors reviewed different theoretical approaches of analysis of inhomogeneous media, including such defects as transition layers, thickness nonuniformity, boundary roughness, etc. Another example of a complex analysis of a dispersive medium based on measurements using spectroscopic ellipsometry was presented by Wong et al. in [14]. Using a two-layer model for homogeneous isotropic ZrON and an interface layer, it was possible to draw conclusions about the oxidation/nitridation of the Zr metal at elevated temperatures. In the present work, the method of analysis of the optical properties of the thin film prepared in the form of a mixture of $TiO_2$ and $Cu_2O/CuO$ with a gradient profile of Cu and Ti elements versus the thickness of the thin film (namely, the gradient $(Ti,Cu)Ox$ thin film) has been presented. Both types of base oxides are well-known to a broad community and are used in many practical applications. Mixtures of copper and titanium oxides have also been shown to have interesting properties that, among other things, depend on the concentration of Cu in addition to $TiO_2$. For example, Horzum conducted a study on the influence of the amount of Cu on the optical and structural properties of thin films of copper–titanium oxide [15]. Furthermore, Mor et al. [16] showed that for a low copper concentration, the prepared

copper–titanium oxide film exhibited n-type electrical conduction, while increasing the copper addition resulted in p-type conduction.

Our previous work [17] has shown that the preparation of a gradient thin film (Ti,Cu)Ox, as opposed to the conventional multilayer stack design, is an interesting alternative to obtain the resistive switching effect.

The optical properties were analysed using reverse engineering. Investigations showed that the analysis of the optical properties of the prepared gradient thin film was complex and required taking into account not only the heterogeneity in the material composition but also the microstructure evolution versus the prepared gradient thin film thickness.

## 2. Experimental

Gradient (Ti,Cu)-oxide thin films were prepared using a home built multi-magnetron sputtering system (Division of Thin Film Technology, Wroclaw, Poland), where three magnetrons arranged in a confocal configuration were used at the same time. In particular, two magnetrons were equipped with Ti (99.95%, Kurt J. Lesker Company, Dresden, Germany) and one was equipped with a Cu (99.95%, Kurt J. Lesker Company, Dresden, Germany) circular disc (28 mm in diameter and 3 mm thick). The targets were co-sputtered in a mixture of oxygen and argon gasses. Each magnetron was powered with a separate 2 kW pulsed dc power supply (Dora Power System) working in unipolar mode. The power delivered to the magnetrons was controlled by the time-width of the groups of pulses ($\tau$) to the duty cycle time $\Delta t$ that can be regulated from 1 ms to 1 s (the pulse width modulation coefficient $pwm = \tau/\Delta t$). To obtain a gradient change in the material composition of the deposited thin films, magnetrons with Ti targets were powered with a constant power coefficient $pwm_{Ti} = 100\%$, while the magnetron with the Cu target was powered with a power that increased linearly during the deposition process according to changes in the $pwm_{Cu}$ coefficient from 0% to 30%. Details of the deposition process have been presented in [18,19]. Thin films were simultaneously deposited on three kinds of substrates: Si, SiO$_2$, and Ti4Al6V, in order to allow investigation of the properties of deposited films applying different research methods. The target–substrate distance was equal to 160 mm.

The deposition time was 240 min, resulting in a total thickness of the deposited film of approximately 340 nm. The average elemental composition of the gradient thin films was measured using an energy dispersive spectrometer (EDAX, Pleasanton, CA, USA) attached to a field-emission FEI Nova NanoSEM 230 scanning electron microscope (SEM, FEI, Hilsboro, OR, USA). XRD (PANalytical, Malvern, UK) and Raman spectroscopy (Thermo Fisher Scientific, Waltham, MA, USA) were used to assess the structural properties of the deposited thin films. Diffraction patterns were obtained with the use of a PANalytical EmpyreanPIXel3D diffractometer with a Cu lamp (40 kV, 30 mA). Moreover, Raman spectra averaged from 10 scans were obtained with a Thermo Scientific DXR™ Raman Microscope in the range of 100 to 700 cm$^{-1}$ with a 455 nm laser diode.

For detailed microstructure investigations of the prepared thin films, a TECNAI G2 FEG Super-Twin transmission electron microscope (TEM, FEI, Hilsboro, OR, USA) with an acceleration voltage of 200 kV was used. For a TEM analysis, thin lamellas were prepared with the FIB Quanta 3D SEM system (FEI, Hilsboro, OR, USA). EDS (EDAX, Pleasanton, CA, USA) was used during TEM measurements to analyse the elemental composition of the gradient thin film as a function of its depth.

The optical properties were analysed by measuring characteristics of the light transmission and reflection coefficients in the wavelength range of 300 to 1100 nm. A scientific grade QE65000 optical spectrophotometer (OceanOptics, Largo, FL, USA) and a coupled deuterium-halogen lamp (OceanOptics, Largo, FL, USA) as a light source were used. The optical properties analysis was performed using the SCOUT software version. 4.17 (Wolfgang Theiss, Akwizgran, Germany) [20].

### 3. Results and Discussion

#### 3.1. Elemental Composition and Structural Properties

Investigations of elemental composition performed with the EDS indicated 84 at. % of copper and 16 at. % of titanium in the whole volume of the prepared thin film—Figure 1a. Furthermore, detailed investigations of the distribution of particular elements at the cross-section of the (Ti,Cu)Ox thin film were performed using TEM-EDS. The normalised concentration profile of Cu versus the thickness of the prepared thin film with the TEM cross-section of the film as a background is presented in Figure 1b. As one can see, the copper concentration increased rapidly with the first 25–30 nm of thin film growth. After that, the concentration of copper tends to saturate and the amount of copper increases much slower up to the sample surface. The best fit to the experimental data of the Cu element distribution was achieved using the exponential growth function that is indicated with a solid line in Figure 1b.

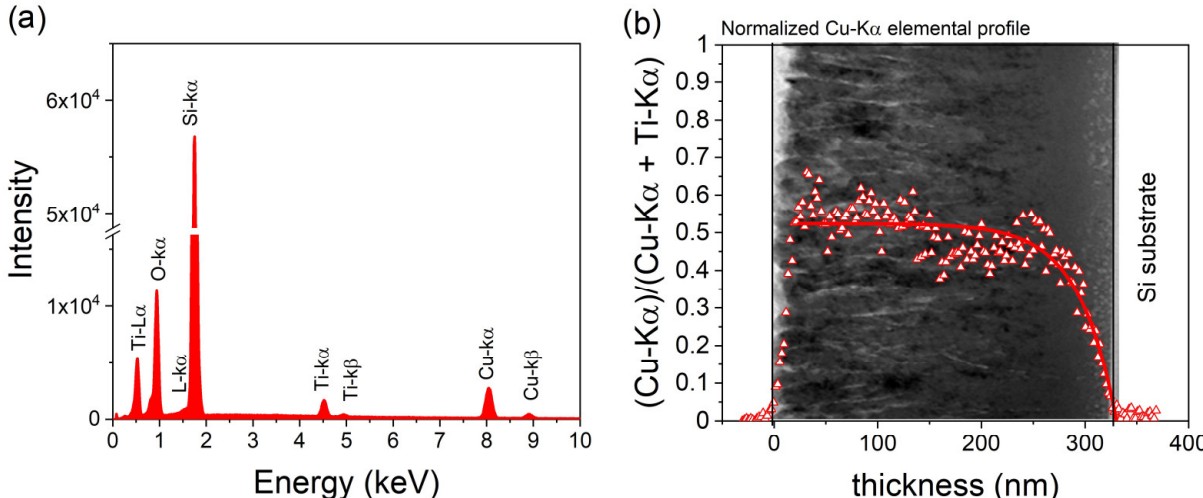

**Figure 1.** (**a**) EDS spectrum of the volume and TEM image (**b**) of the cross-section together with elemental distribution of Cu measured using EDS X-ray microprobe of deposited gradient (Ti,Cu)Ox thin film. Symbols—experimental data, solid line—fitted theoretical curve.

The elemental composition profile in the cross-section of the prepared thin film (Figure 1a) has been calculated using the intensity of the Cu-K$\alpha$ line to the sum of the Cu-K$\alpha$ and Ti-K$\alpha$ lines measured with EDS (Figure 1b). The obtained result confirms the intentionally increased sputtering power supplied to the magnetron with the Cu target. The deviation of the obtained Cu concentration profile from the preset $pwm_{Cu}$ factor profile is due to a different distribution of the power released in the Cu target in relation to changes in the $pwm_{Cu}$ parameter during the deposition process. A more extensive discussion of this subject is presented by the authors in [17].

The microstructure of the deposited thin film was investigated using XRD and Raman spectroscopy. X-ray diffraction measurements showed that the gradient coatings deposited were rather amorphous, however, in the 2θ range of ca. 32° to 40° wide and low intensity peaks can be distinguished (Figure 2a). They might be related to the (002), (111), or (200) planes of CuO-monoclinic phase [21]. Such low intensity of the peaks can be related to the low sensitivity of the used XRD diffractometer to detect fine nanocrystallites or a large amount of the amorphous phase in the prepared (Ti,Cu)Ox thin film. The Raman spectrum of the gradient thin film is presented in Figure 2b. Analysis revealed peaks that are related to the CuO and Cu$_2$O phases. The Raman spectrum consists of three main one-phonon CuO modes, i.e., the most intense peak at 285 cm$^{-1}$ assigned to the Ag mode and peaks at 345 and 620 cm$^{-1}$ assigned to the Bg modes [22,23]. The presence of Cu$_2$O nanocrystals is testified mainly by the occurrence of the second-order Raman allowed mode peaks at ca. 213 and 560 cm$^{-1}$ [23,24]. The crystalline phases of titanium dioxide were not present

in the XRD or Raman spectra, showing that they occurred in the amorphous phase in the deposited thin films.

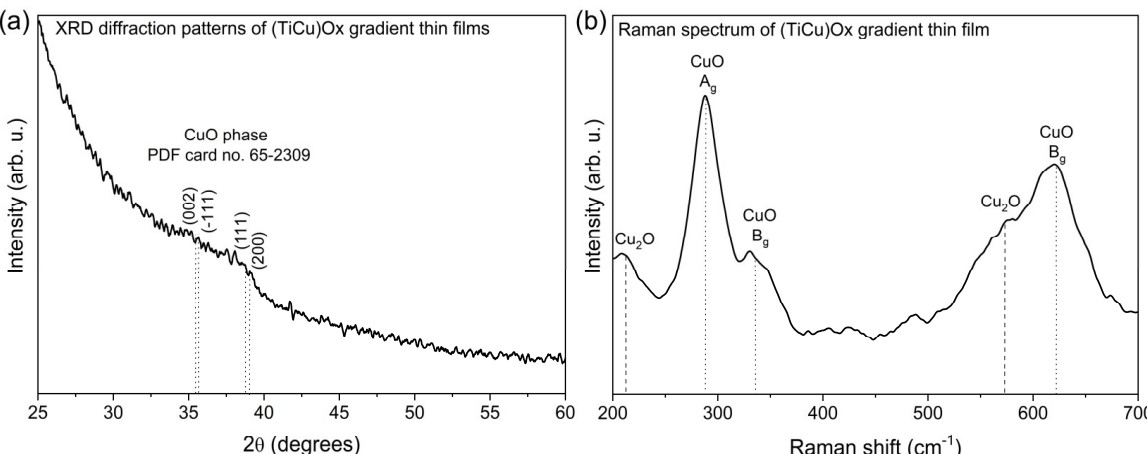

**Figure 2.** The XRD (**a**) and Raman spectroscopy (**b**) results of analysis of microstructure properties of prepared (Ti,Cu)Ox gradient thin film.

### 3.2. Analysis of Optical Properties

The results of the investigations of optical properties of the prepared thin film deposited on silica substrate showed that the thin film was semi-transparent in the visible part of the optical radiation (symbols in Figure 3). The optical band gap for the allowed indirect transitions was estimated at approximately 1.36 eV from the transmittance spectrum using the Tauc formulae [17]. Such a relatively narrow band gap, compared to pure $TiO_2$ (3.2÷3.4 eV), results from a high concentration of copper in the prepared thin film, and from the presence of cupric ($CuO$) and cuprous ($Cu_2O$) oxides for which the band gap width is usually reported to be in the range of 1.21÷1.51 eV, e.g., [25] and 2.1÷2.6 eV, e.g., [26], respectively.

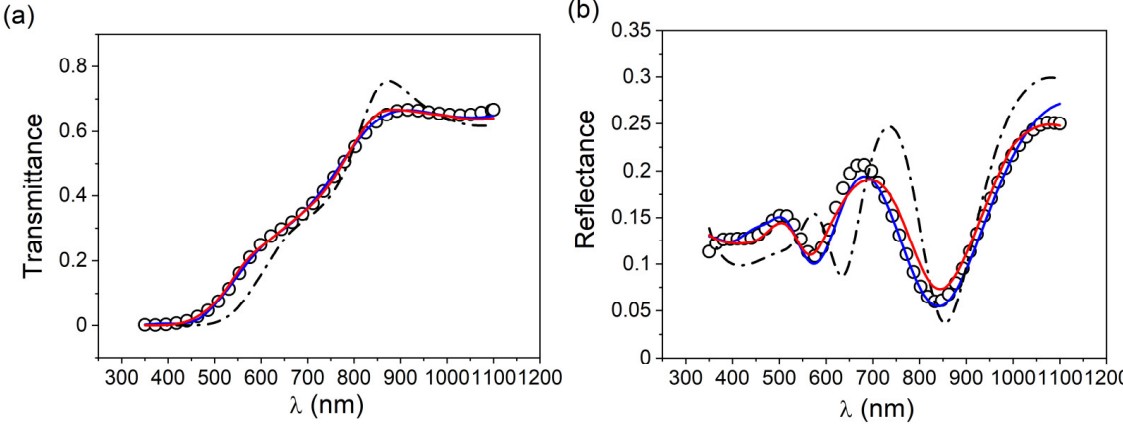

**Figure 3.** Experimental (symbols) and theoretical (lines) characteristics of (**a**) optical transmittance and (**b**) reflectance of gradient (Ti,Cu)Ox thin film coating. Theoretical lines: blue solid line single homogenous oxide model; black dash-dot line for composition profile EMA model; and red solid line for composition profile EMA including porous-void microstructure model.

Further analysis of optical properties was performed using the reverse engineering method. The simplest way to analyse the optical properties of the thin film coating is assuming a single homogeneous oxide model. In this present study, in theoretical calculations, the dielectric background, together with oscillators according to functions derived by Tauc–Lorentz [27], Kim [28], and O'Leary–Johnson–Lim [29], were used. Figure 3 shows the course of the measured experimental characteristics of the light transmission and reflection coefficients (symbols) together with the theoretical curves (solid blue lines) obtained

as the best fit to the experimental data. The details of the function parameters used are given in Table 1.

**Table 1.** Details of the function parameters used in single homogenous oxide model of investigated gradient (Ti,Cu)Ox thin film.

| Dielectric Background | Real Part | | Imaginary Part | |
|---|---|---|---|---|
| | 0.0000015 | | 0.0 | |
| Tauc–Lorentz | Resonance frequency ($cm^{-1}$) | Oscillator strength | Damping ($cm^{-1}$) | Gap energy ($cm^{-1}$) |
| | 4,320,035 | 604,104.250 | 24,296.500 | 5995.517 |
| O'Leary–Jonhson–Lim | Oscillator strength | Gap energy ($cm^{-1}$) | Gamma ($cm^{-1}$) | Decay ($cm^{-1}$) |
| | 42.166 | 4,721,061.500 | 150.000 | 141,032.800 |
| Kim | Resonance frequency ($cm^{-1}$) | Oscillator strength | Damping ($cm^{-1}$) | Gaus-Lorentz-Switch |
| | 22,828.300 | 8476.600 | 4433.112 | 0.001 |

The analysis performed shows that the application of a properly selected model for a single homogeneous oxide film gives the possibility of obtaining a very good fit of the simulated curves to the experimental data. The deviation factor (goodness of fit) from the measured experimental data was dev = $1.243 \times 10^{-4}$ (Pearson coefficient factors: $\rho_T = 0.99967$ for transmittance and $\rho_R = 0.98928$ for reflectance). The calculated spectra could be further used to derive the real (*n*) and imaginary (*k*) parts of the complex refractive index of the prepared coating. However, interpretation of the *n* and *k* spectra calculated in this way would be difficult, since the values obtained represent some kind of 'equivalent' of the whole thin film, ignoring the fact of changes in the composition of the material versus its thickness.

Therefore, further analysis was performed, taking into account that the prepared thin film (Ti,Cu)Ox (1) was a mixture of particular oxides, namely $TiO_2$, $CuO$, and $Cu_2O$, and (2) particular elements, namely Cu and Ti, were distributed versus the thickness of the thin film according to some profile specified, as indicated in Figure 1b. For calculations of theoretical transmittance and reflectance spectra, the dielectric functions available in the SCOUT software database were used. Furthermore, in the construction of the prepared thin film, the surface roughness visible in Figure 1b was included in the analysis to achieve a better fit to the experimental data.

Figure 4 presents a schematic representation of the two assumed parts of the evaluated layer stack model for the prepared (Ti,Cu)Ox thin film, that is, the part representing the rough surface and the thin film with a concentration gradient of the $Cu_xO$ component.

A top sublayer was about 32 nm thick and was a mixture modelled using EMA of the composite $Cu_xO$ layer and air with the corresponding EMA volume fraction factors of 0.694 and 0.306, respectively. Additionally, the $Cu_xO$ composition was calculated using EMA as a mixture of $Cu_2O$ and $CuO$. The best match between theoretical and experimental spectra was obtained assuming the absence of a $TiO_2$ component in the near-surface region of the coating.

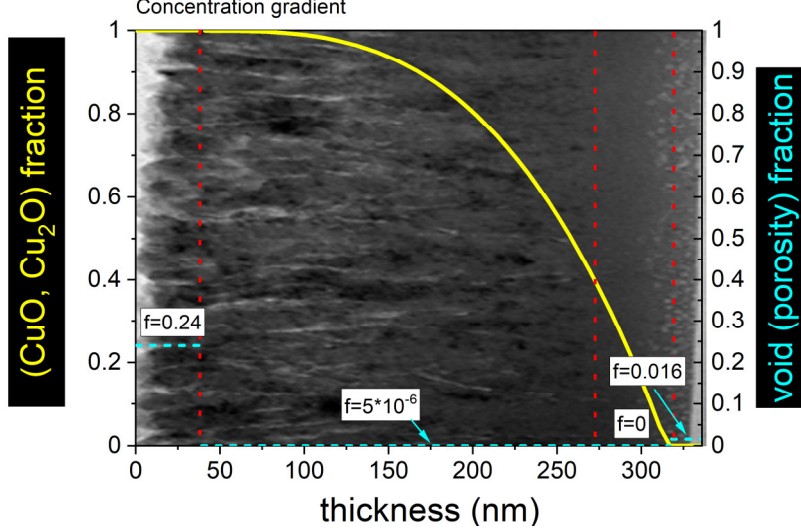

Rough surface $\begin{cases} Cu_xO \begin{cases} f_{CuO} = 0.66 \\ f_{Cu_2O} = 0.34 \end{cases} \\ f_{Air} = 0.306 \end{cases}$

Concentration gadient single film $\begin{cases} f_{TiO_2} = 1 - f_{Cu_xO} \\ f_{Cu_xO} = A - Bx^3 \end{cases} \begin{cases} f_{Cu_2O} = 0.813 \\ f_{CuO} = 0.187 \end{cases}$

**Figure 4.** Schematic representation of the construction of the parts of equivalent layer stack model of the prepared (Ti,Cu)Ox thin film used for the simulations of the transmittance and reflectance spectra, containing the rough surface and the volume of prepared thin film with concentration gradient of the $Cu_xO$ component. $f$—volume fraction factor in EMA model and $A$, $B$—are the gradient function parameters.

The main volume of the thin film was calculated assuming $TiO_2$ as a base oxide, in which the concentration of the $Cu_xO$ volume fraction factor changed according to the mathematical function that allowed one to obtain the best approximation of the theoretical transmittance and reflectance curves calculated to the experimental ones. The best fit (dev = $3.38 \cdot 10^{-3}$, $\rho_T = 0.99054$, and $\rho_R = 0.91255$) was obtained for the gradient profile of the volume fraction factor for the Cu–oxide components described with the shape function $f_{Cu_xO} = A - Bx^3$, where $A$ and $B$ are the function parameters and $x$ is the relative depth versus the total thickness. Detailed values of the parameters $A$, $B$ are given in Table 1. The shape of the calculated profile of the $Cu_xO$ component versus the thickness of the thin film is presented in Figure 5.

**Figure 5.** Concentration gradient profile of $Cu_xO$ components calculated for the prepared (Ti,Cu)Ox thin film coating together with values of the calculated porosity/void fraction coefficient ($f$).

It should be noted that the calculated concentration profile of the mixture of CuO and $Cu_2O$ components (Figure 5) is in very good agreement with the shape of the curve that fits very well with the experimental data of Cu concentration revealed with energy dispersive spectroscopy analysis (Figure 1b). The $Cu_xO$ component in the body is mainly composed of $Cu_2O$, volume fraction factor $f_{Cu_2O} = 0.813$, whereas the surface area was rich with the

CuO ($f_{CuO} = 0.66$). The presence of both crystal phases of copper oxides was confirmed by Raman spectroscopy investigations (Figure 2b). The dominant presence of the CuO phase on the surface of the coating is fully understandable, since the formed surface layer of the film can easily oxidise.

The presented calculations for the heterogeneous concentration gradient thin film did not reveal a satisfactory solution of the calculated theoretical curves with respect to the experimental data, black dashed-dotted lines in Figure 3. This fact may indicate that the composition of the thin film is not the only factor responsible for the formation of the transmittance/reflectance spectra. Therefore, the microstructure of the prepared thin film was taken into account in the analysis as another possible factor.

Detailed analysis of the TEM image presented in Figure 6 yields a strong conclusion about the heterogeneity also in the microstructure in the prepared gradient thin film. The film consisted of at least four areas with different types of microstructures. Directly near the substrate, the film was amorphous (1st area—about 20 nm wide) with a fairly significant number of voids, after which it turned into a fairly dense one (2nd area—30 nm ÷ 40 nm wide). At a distance of 50 nm ÷ 60 nm from the substrate, a crystalline region with a fibrous–porous structure is visible (3rd area about 260 nm wide). The surface of the film (4th area) is strongly diversified with visible porosity. Furthermore, detailed structure analysis performed using bright-field and high-resolution transmission electron microscopy images confirmed the amorphous behaviour of the thin film at the beginning of its growth and the presence of copper oxide phases, that is, $Cu_2O$ and CuO in the fibrous crystalline area [9].

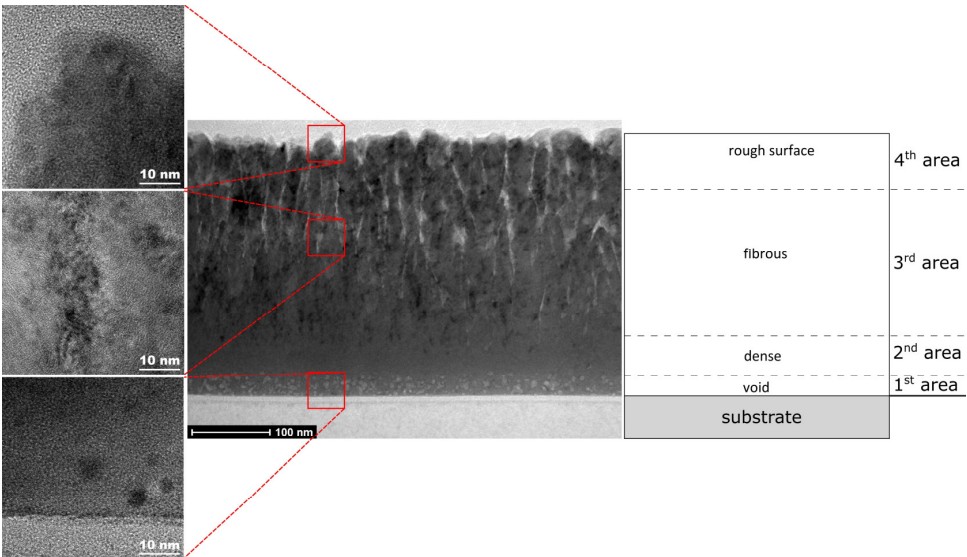

**Figure 6.** TEM cross-section analysis of prepared (Ti,Cu)Ox thin film.

Therefore, in addition, in the calculation of the theoretical transmittance and reflectance spectra, a combination of the solid material with an air (as a void or pore material) was used to model the changes in the microstructure of the prepared gradient thin film. Figure 7 presents an elaborated equivalent layer stack model used for each subarea, which allowed to obtain the best fit (dev = $2.214 \times 10^{-4}$, $\rho_T = 0.99929$, $\rho_R = 0.97908$) of calculated theoretical transmittance and reflectance spectra to the experimental ones (the red solid lines in Figure 3). As can be seen, taking into account the heterogeneity of the prepared thin film not only in the elemental composition but also in the microstructural properties of the prepared coating allowed a much better agreement with the experimental spectra. For surface roughness, again the EMA model with a constant air volume fraction factor was used. The three remaining areas were modelled using EMA combined with a concentration gradient, where the parameters of the void/porosity factors (f) for each subarea were calculated independently. The f values that resulted in the best fit obtained from the calculated transmittance and reflectance are included in Figure 5. As one can see, in

general, the calculated porosity profile for the studied thin film agrees well with the TEM observations. As expected, the most porous part of the film was the surface, for which the calculated volume fraction factor $f_{Air}$ was equal to 0.24. The contribution of voids in the thin film microstructure in the area near the substrate was calculated as $f_{voids} = 0.016$. Within the fibrous crystalline area, the $f$ factor was equal to $5 \cdot 10^{-6}$. The most dense part of the film is the area located between 30 and 70 nm from the substrate, which is characterised with a constant profile of a fraction factor $f$ equal to zero. Particular steps of the analysis performed and parameters used are summarised in Table 2.

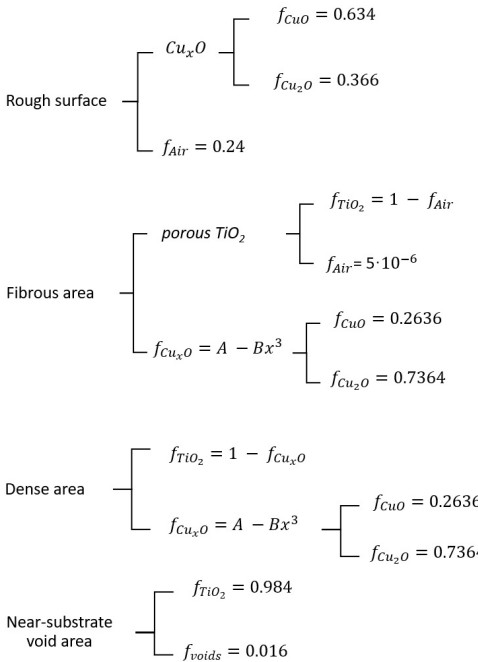

**Figure 7.** Schematic representation of the construction of the parts of equivalent layer stack model of the prepared gradient (Ti,Cu)Ox thin film used for the simulations of the transmission and reflection spectra, containing four areas with different microstructure properties as marked in Figure 6. $f$—volume fraction factor in EMA model, $A$, $B$—are the function parameters.

**Table 2.** Summary of models and parameters used at each step of performed analysis.

| No. | Layer Stack Model | Layer Thickness (nm) | No of Sublayers | Parameters Determined | Deviation Factor |
|---|---|---|---|---|---|
| 1 | single homogenous oxide film | 363.6 | 1 | $n, k, d$ | 0.0001243 |
| 2 | rough surface | 38 | 1 | $f_{Air} = 0.306; f_{Cu_xO} : \begin{matrix} f_{Cu_2O} = 0.34 \\ f_{CuO} = 0.66 \end{matrix}$ | 0.0033795 |
| | single gradient concentration film | 300 | 32 | $f_{TiO_2} = 1 - f_{Cu_xO}, f_{Cu_xO} = A - Bx^3, A = 1, B = 1.247$ $f_{Cu_xO} : \begin{matrix} f_{Cu_2O} = 0.813 \\ f_{CuO} = 0.187 \end{matrix}$ | |
| 3 | rough surface | 39 | 3 | $f_{Air} = 0.24; f_{Cu_xO} : \begin{matrix} f_{Cu_2O} = 0.366 \\ f_{CuO} = 0.634 \end{matrix}$ | 0.0002214 |
| | fibrous area | 233 | 20 | $f_{Cu_xO} = A - Bx^3, A = 0.7925, B = 5.079$ $porous TiO_2 : \begin{matrix} f_{TiO_2} = 1 - f_{voids} \\ f_{voids} = 5 \cdot 10^{-5} \end{matrix}$ | |
| | dense area | 45 | 6 | $f_{void} = 0; f_{TiO_2} = 1 - f_{Cu_xO} f_{Cu_xO} = A - Bx^3, A = 0.3716, B = 0.0386$ $f_{Cu_xO} : \begin{matrix} f_{Cu_2O} = 0.7364 \\ f_{CuO} = 0.2636 \end{matrix}$ | |
| | near-substrate void area | 19 | 3 | $f_{voids} = 0.016, f_{TiO_2} = 0.984$ | |

## 4. Conclusions

The method of analysis of a graded thin film of mixed titanium and copper oxides with Cu concentration, varied as a function of the prepared thin film thickness, has been presented. Performed analysis has shown that the shape of the concentration profile of the Cu components calculated on the basis of optical spectra using EMA, combined with the concentration gradient of the volume fraction factor, was very similar to the experimental one obtained using the EDS. Moreover, consideration of the microstructure properties in the analysis allowed to obtain a satisfying match between the theoretical light transmission and reflection characteristics and experimental data for the prepared composite gradient thin film. The analysis allowed us to calculate the profile of the volume fraction factor that could be considered as a factor representing the 'porosity' in the prepared thin film.

**Author Contributions:** Conceptualization, J.D.; methodology, J.D., M.M. and D.W.; formal analysis, J.D., M.M. and D.W.; investigation, J.D., M.M., D.W., E.M., A.W. and P.C.; resources, J.D.; data curation, M.M. and E.M.; writing—original draft preparation J.D., M.M., E.M. and D.W.; writing—review and editing, J.D., M.M. and D.W.; visualization, J.D.; supervision, J.D.; project administration, J.D.; funding acquisition, J.D. All authors have read and agreed to the published version of the manuscript.

**Funding:** This work was financed from the sources given by the Polish National Science Centre (NCN) as a research project number DEC-2018/29/B/ST8/00548 in the years 2019–2022.

**Institutional Review Board Statement:** Not applicable.

**Informed Consent Statement:** Not applicable.

**Data Availability Statement:** Data underlying the results presented in this paper are not publicly available at this time but may be obtained from the authors upon reasonable request.

**Conflicts of Interest:** The authors declare no conflict of interest.

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
