# Peer review of "Reverse Engineering Analysis of Optical Properties of (Ti,Cu)Ox Gradient Thin Film Coating"

_coatings, doi:10.3390/coatings13061012_

Round 1

Reviewer 1 Report

For the manuscript entitles "Reverse engineering analysis of optical properties of (Ti,Cu)Ox gradient thin film coating",authors report the study for the gradient (Ti,Cu)Ox film. Some representations and discussions in the manuscript can be improved.

1. (line 185-186) Please show the procedure for determined the optical bandgap quantity 1.36eV.

 2. Please explain the reasons for choosing the mathematics form A-Bx^3(line 242-line 243). The unit of B should be described.

3. As a pre-selected model in this work with very good fitting result (line 205), the fitting parameters and references for this "simple single oxide model" should be well described.

4. Is "simple single oxide model" (line 193-line194) and "model for a single homogeneous film"(line 205) the same? If so, please use the same name for 

model description in this paper.

5. Please show references for the described bandgaps of TiO2 (line 187), CuO and Cu2O (line 189). 

6. Please show references for the signal(s) position identification of XRD (line 167) and Raman spectra (line 174-176).

Author Response

Dear Reviewer,
We would like to express our gratitude for the comments that allowed us to improve our manuscript. We have included them in the revised version of our article. Responses to the reviewer's questions are posted in the attached file.

Reviewer 2 Report

Journal: Coatings

Manuscript ID: coatings-2419248

Manuscript Type: Research Article

Manuscript Title: Reverse engineering analysis of optical properties of (Ti,Cu)Ox gradient thin film coating

Comments:

1.     This is an interesting work on reverse engineering analysis of optical properties of Ti-Cu-Oxide gradient thin film coating using reactive co-sputtering method.  

2.     In this work, the oxide material properties and performance, i.e. physical, optical and microstructural properties has been extensively discussed and elaborated so that the future researchers and readers will know what performance values are to be expected from this kind of oxide. From a quick search, there is a quite recent and relevant research paper that could be a reference for this work, i.e.

(i)               “Investigation of the structural and optical properties of copper-titanium oxide thin films produced by changing the amount of copper”, Thin Solid Film, Vol 685, (2019), pp. 293 – 298. DOI: 10.1016/j.tsf.2019.06.052

(ii)             “Physical and dispersive optical characteristics of ZrON/Si thin-film system”, Applied Physics A: Materials Science and Processing, 115 (2014), pp. 1069 – 1072. DOI: 10.1007/s00339-013-7947-1

for coherent arrangement of introduction section as well as results and discussion. The relevant information added will be useful for readers who are not familiar with oxides, especially on the optical and microstructural properties.

3.     The grammar in the manuscript should be revised. Sentence structure is also a concern in this manuscript. I suggest the authors to check the English language and grammar by native English speakers.

4.     Although the authors have conducted quite several characterizations for this study and the manuscript contains major essential results, however, based on the evaluation above, this work still requires a minor revision to become publishable in the Coatings journal.

Moderate editing of English language required. 

Author Response

(The authors gave the same response as above.)
